# Analysis of Global Microbial Safety Incidents in Frozen Beverages from 2015 to 2024

**DOI:** 10.3390/foods14183238

**Published:** 2025-09-18

**Authors:** Yulong Qin, Wenbo Li, Zhaohuan Zhang, Yuying Lu, Gui Fu, Xu Wang

**Affiliations:** 1College of Food Science and Technology, Shanghai Ocean University, Shanghai 201306, China; qyl1218@139.com (Y.Q.); lwb499520@163.com (W.L.); zh-zhang@shou.edu.cn (Z.Z.); 19816295951@163.com (Y.L.); d240400113@st.shou.edu.cn (G.F.); 2International Research Center for Food and Health, Shanghai Ocean University, Shanghai 201306, China; 3National Center of Technology Innovation for Dairy, Hohhot 010110, China; 4Inner Mongolia Yili Industrial Group Co., Ltd., Hohhot 010110, China

**Keywords:** frozen beverages, ice cream, microbial contamination risks, foodborne pathogens, risk monitoring

## Abstract

Microbial contamination in frozen beverages threatens public safety and food quality. By systematically analyzing safety incidents, potential microbial hazards can be identified. This study reviewed 155 microbial safety incidents related to frozen beverages reported by nine international regulatory agencies from January 2015 to December 2024, as well as 903 incidents published by the State Administration for Market Regulation of China. The results indicate a higher risk in Southeast Asia, particularly in Malaysia (16.13%) and Thailand (11.61%). In China, the risks are concentrated in South China (Guangdong, 14.52%), Northeast China (Liaoning, 10.20%; Heilongjiang, 9.87%), and the Huang-Huai-Hai region (Henan 6.87%; Shandong 5.99%). Statistical analysis reveals that non-compliance incidents related to coliforms account for 67.7% globally, while incidents involving pathogens such as *Listeria monocytogenes*, *Staphylococcus aureus*, *Salmonella*, and Norovirus account for 6.4%. The characteristics in the Chinese market align with global trends, with the highest proportion of coliform exceedance (41%), while the incidence of pathogenic contamination remains relatively low (0.6%). Further analysis of different types of frozen beverages (ice cream, ice milk, ice frost, ice lolly, sweet ice, edible ice, and others) and their association with microbial hazards reveals significant issues with ice cream products globally; however, in the Chinese market, the contamination problems with ice milk and ice lolly are more severe. This study provides regional and category-specific data for the microbial risk assessment of frozen beverages and offers guidance for regulatory agencies and enterprises to implement targeted control measures, including optimizing sampling plans, enhancing hygiene controls during production processes, and promoting compliance in cold chain management. Consequently, this approach effectively reduces the risk of foodborne diseases and enhances the overall safety level of the industry, demonstrating significant practical application value and public health significance.

## 1. Introduction

Frozen beverages occupy a significant position in the global food consumption sector. The ice cream alone is projected to continue expanding at a compound annual growth rate of 4.2%, with the overall market size expected to reach $91.9 billion by 2027 [1]. However, the rich nutritional content of frozen beverages [2], their neutral pH environment, and long-term storage characteristics provide potential conditions for microbial growth [3]. In addition to traditional hygiene risks such as coliforms and total plate counts, frozen beverages are associated with contamination from various foodborne pathogens, primarily including *Salmonella* spp. [4,5], *Listeria monocytogenes* [6], *Bacillus cereus* [7], Shiga toxin-producing *Escherichia coli* [8], and *Staphylococcus aureus* [9]. The microbial risks pose a serious threat to the food safety of frozen beverages and have become a key factor restricting the development of this industry. It is crucial to address food safety issues in this sector with high priority and to implement effective measures to resolve them.

By conducting an in-depth search and systematic review of official regulatory agency databases at the national level, and relying on comprehensive, long-term, and highly credible data from national official regulatory databases, retrospective studies can more accurately identify potential sources of risk. This provides a data foundation for the hazard identification phase in risk assessment and offers empirical support and scientific basis for decision-making based on big data [10]. To identify emerging food safety risks early, Kleter et al. employed statistical analysis methods to comprehensively review the alert information released by the European Commission’s Rapid Alert System for Food and Feed (RASFF) from July 2003 to June 2007 [11], focusing particularly on the potential links between different product types and specific hazard categories, including microbial and chemical risks. Additionally, the study analyzed the origin information of related products or the countries/regions submitting alert notifications to explore the geographical distribution characteristics of contamination issues and the risks of cross-border transmission within the food supply chain. Meanwhile, Banach et al. conducted a systematic review and assessment of microbial and chemical contaminants in spices and herbs by retrieving alerts and monitoring records from multiple authoritative data sources, including the RASFF database and the World Health Organization’s Global Environmental Monitoring System (GEMS) database, aiming to uncover potential food safety risk points and extract relevant specific data as key input parameters for risk assessment [12]. Recently, a study focused on 2713 seafood-related data entries published in the RASFF database between 2011 and 2015, delving into the origin information of seafood, the hazard categories involved, and their potential correlations, while discussing the specific impacts of these factors on seafood safety and market circulation [13]. The in-depth exploration of key and valuable data within core databases has become an indispensable part of the risk assessment process, providing robust and up-to-date evidence for the precise identification of risks and hazards [14]. However, there is currently a lack of systematic analysis of mainstream and accessible government alerts and monitoring data on microbial risks associated with frozen beverages on a global scale to support the risk assessment of hazards in frozen beverages. This issue may stem from several aspects: First, frozen beverages are often mistakenly believed to be completely free of microbial growth due to the low-temperature environment, leading to an underestimation of their associated risks. Second, relevant safety data is dispersed across multiple regulatory systems with inconsistent standards, making it challenging to conduct global or cross-regional integrative analyses. Furthermore, existing research predominantly focuses on broad categories such as dairy products or cold chain foods, lacking in-depth exploration of the microbial contamination characteristics specific to subcategories of frozen beverages.

Therefore, this study systematically collected and analyzed 155 records of microbial safety incidents related to the import and export of frozen beverages from nine authoritative official agencies worldwide between 2015 and 2024. Furthermore, for the Chinese market, this study collected and analyzed 903 microbial safety incidents published by the State Administration for Market Regulation of China. A comprehensive multidimensional statistical analysis was conducted, covering the origin information of risk events, statistics on microbial non-compliance in risk events, and risk information statistics corresponding to different types of frozen beverages. The aim is to reveal the potential safety risks of frozen beverages globally, provide a data foundation for risk assessment, and guide targeted control strategies for potential hazards. The research findings will be directly applied to quality and safety control practices within the frozen beverage industry chain, significantly enhancing regulatory efficiency, promoting compliance in international trade, and safeguarding consumer health rights.

## 2. Materials and Methods

### 2.1. Data Collection on Microbial Safety Incidents of Frozen Beverages in Global Import and Export Trade

Based on the PRISMA guidelines (https://www.prisma-statement.org (accessed on 12 September 2025)), we conducted a data collection on microbial safety incidents related to frozen beverages in global import and export trade. This study conducted a search using keywords related to frozen beverages and microorganisms. Based on the Chinese national standard GB/T 30590-2014 [15] for the classification of frozen beverages, we categorized ice cream, ice milk, ice frost, ice lolly, sweet ice, and edible ice. Relevant search terms included, but were not limited to frozen beverages, microorganisms, bacteria, total plate count, foodborne pathogens, coliforms, *E. coli*, *Listeria*, *L. monocytogenes*, *Salmonella*, *S. aureus*, Norovirus. We collected data on microbial incidents related to frozen beverages in import and export inspections published by official entities of major trading countries worldwide from January 2015 to December 2024.

### 2.2. Criteria for Data Inclusion and Exclusion

The inclusion criteria are as follows: the data must include product type, notification date, and details of microbial non-compliance, as well as the country of origin. If any necessary information is missing from the incident data, that entry will be excluded. Given the extensive nature of international trade, we will conduct a thorough comparison of the producers and product names in the incident data to ensure the identification of potential duplicate reports across regulatory agencies, thereby avoiding double counting. The data primarily comes from Table 1, shown below.

### 2.3. Data Collection on Microbial Safety Incidents of Frozen Beverages in the Chinese Market

This study conducted a search using keywords related to frozen beverages and microorganisms. Based on the Chinese national standard GB/T 30590-2014 for the classification of frozen beverages, we categorized ice cream, ice milk, ice frost, ice lolly, sweet ice, and edible ice. Relevant search terms included, but were not limited to, frozen beverages, microorganisms, bacteria, total plate count, foodborne pathogens, coliforms, *E. coli*, *Listeria*, *L. monocytogenes*, *Salmonella*, *S. aureus*. We collected data on incidents of non-compliance in the safety supervision and spot checks of frozen beverages in the domestic market, as reported by official Chinese agencies from January 2015 to December 2024. The inclusion criteria are the same as those for microbial safety incidents related to frozen beverages in global import and export trade.

The primary data sources were the State Administration for Market Regulation of China and the Market Supervision Administration of various provinces in China. This study conducts a comprehensive comparison of producer and product names within the event data to ensure that there are no duplicate reports. This article does not provide specific links.

### 2.4. Geographic Information Visualization Analysis of Microbial Safety Incidents in Frozen Beverages

This study conducts an in-depth analysis of the geographical information contained within the microbial risk data collected from frozen beverages. Utilizing the open-source visualization chart library ECharts (v6.0.0) based on JavaScript [16], we created a heat map illustrating the distribution of non-compliant microbial indicators in imported and exported frozen beverages from major trading countries worldwide between 2015 and 2024. Additionally, by integrating announcement information released by the State Administration for Market Regulation of China and various provincial market supervision bureaus, we also generated a heat map depicting the origins of non-compliant products identified during safety supervision sampling inspections of frozen beverages across different provinces in China during the same period.

### 2.5. Analysis of the Contamination Spectrum of Microbial Safety Incidents in Frozen Beverages

This study focuses on the microbial risk data of frozen beverages imported and exported by major global trading countries from 2015 to 2024, as well as the safety supervision and sampling inspection data of frozen beverages in the Chinese market. The analysis categorizes the data based on non-compliance with microbial indicators, which include excessive coliforms, combined exceedance of coliforms and total plate count, excessive total plate count, and exceedance/detection of pathogenic microorganisms. The standard for non-compliance is determined by the legal standards established by the country where the non-compliance incident is detected. For pathogenic microorganisms, specific strains are further classified, primarily including *L. monocytogenes*, *S. aureus*, *Salmonella* spp., and Norovirus. The study utilizes Excel software to statistically calculate the non-compliance items of microbial indicators in frozen beverages and their proportions, and pie charts are generated using Origin 2024.

### 2.6. Analysis of Microbial Safety Incidents in Frozen Beverage

This study categorizes frozen beverages based on the Chinese national standard GB/T 30590-2014 for the classification of frozen beverages and analyzes the microbial risk data associated with these products. The frequency of non-compliance with microbial indicators across different categories of frozen beverages was statistically assessed, including excessive levels of coliforms, combined exceedances of coliforms and total plate count, excessive total plate count, and the presence/exceedance of pathogenic microorganisms. This quantification aims to evaluate the severity of microbial contamination in various categories of frozen beverages. A heatmap illustrating the frequency of non-compliance with microbial indicators across different categories of frozen beverages was generated using the ChiPlot online platform (https://www.chiplot.online/ (accessed on 12 September 2025)). Building on this, a cluster analysis was conducted to explore the similarities and differences in non-compliance with microbial indicators among the various categories of frozen beverages, using the Euclidean Distance Clustering Algorithm.

### 2.7. Statistical Analysis

Associations among attributes were investigated using chi-square test for category comparison by SPSS 20.0 (SPSS Inc., Chicago, IL, USA). Diagrams were prepared by Origin 2021 (OriginLab Corporation, Northampton, MA, USA).

## 3. Results

### 3.1. Screening Results of Microbial Safety Incidents in Frozen Beverages

A total of 1787 microbial safety incident data entries were obtained through database searches, including 237 incidents from various national databases and 1550 incidents from the database of the State Administration for Market Regulation of China database. After data screening, no duplicate statistics were found. A screening based on titles and theme retained 1104 incidents. A full-text re-screening of the incident data was conducted, and based on the inclusion and exclusion criteria, a total of 46 incidents were ultimately excluded (14 incidents where the product type could not be extracted from the incident reports and 32 incidents where the country of origin could not be determined). For a detailed overview of the specific process following the PRISMA method, please refer to Figure 1. The detailed study of the characteristics is presented in Table 2.

### 3.2. Spatiotemporal Distribution of Microbial Safety Incidents in Frozen Beverages Imports and Exports Among Major Global Trading Nations

This study collected reports of non-compliance with microbial indicators during the inspection of imported and exported frozen beverages from major global trading countries between 2015 and 2024, totaling 155 cases. The distribution of these incidents by year is illustrated in Figure 2a. Specifically, the highest number of reported non-compliance incidents occurred in 2015, with 36 cases, while 2018 recorded the lowest, with no related reports throughout the year. The number of non-compliance incidents reported in the other years is as follows: 24 cases in 2016, 9 cases in 2017, 7 cases in 2019, 12 cases in 2020, 18 cases in 2021, 9 cases in 2022, 19 cases in 2023, and 21 cases in 2024. Overall, the occurrence of non-compliance incidents with microbial indicators exhibited a fluctuating trend, indicating that the outbreaks of such non-compliance events are characterized by randomness.

The comparison of microbial safety incidents in different regions of the world is presented in Table 3. The distribution of the countries of origin for the non-compliance incidents is shown in Figure 2b, and percentages are of total global cases (*n* = 155): Malaysia with 25 cases (16.13%), Thailand with 18 cases (11.61%), New Zealand with 17 cases (10.97%), China Taiwan with 15 cases (9.68%), Russia with 15 cases (9.68%), the United States with 13 cases (8.39%), Italy with 8 cases (5.16%), China Hong Kong with 6 cases (3.87%), and China, Singapore, and Turkey each with 4 cases (2.58%). Additionally, Germany and Australia each reported 3 cases (1.94%), while Vietnam, India, South Korea, the Philippines, and France each reported 2 cases (1.29%). Furthermore, there was 1 case each from the United Kingdom, Sudan, Lithuania, Latvia, Kyrgyzstan, and Belgium (0.65% each), and 4 cases with unspecified country of origin (2.58%).

Geographic information analysis indicates a concentrated distribution of non-compliance incidents in the countries of origin, with Southeast Asian countries, including Malaysia and Thailand, exhibiting a higher microbial safety risk for frozen beverages. In contrast, Western European countries displayed lower microbial safety risk characteristics.

### 3.3. Microbial Safety Incident Contamination Spectrum of Frozen Beverages Imports and Exports Among Major Global Trading Nations

The distribution of non-compliance incidents related to microbial indicators in the import and export of frozen beverages among major global trading countries from 2015 to 2024 is illustrated in Figure 3. Among all the reasons for microbial indicator non-compliance, incidents of excessive coliforms accounted for a dominant 67.7%. Coliforms serve as an indicator of fecal contamination in food, and current food and water quality regulations worldwide are primarily based on coliforms concentration. The quantity of coliforms reflects the degree of food contamination and the potential health risks, while also indicating the presence of intestinal pathogenic bacteria [17]. In the context of microbial indicator non-compliance in the trade of frozen beverages, incidents of excessive total plate count constituted 15.6%. Although this figure is lower than that for coliforms, it remains significant. Additionally, incidents of both coliforms and total plate count exceeding standards accounted for 10.3%.

Furthermore, it is noteworthy that the detection of pathogenic microorganisms represented 6.4% of all non-compliance incidents related to microbial indicators in the import and export of frozen beverages. This includes *L. monocytogenes* (2.6%), *S. aureus* (1.9%), *Salmonella* (1.3%), and Norovirus (0.6%). The detection of pathogenic microorganisms is directly linked to the risk of foodborne disease transmission, and the potential threat to consumer health should not be overlooked.

### 3.4. Cluster Analysis of Microbial Safety Incidents in Frozen Beverage Imports and Exports Among Major Global Trading Countries

Based on the microbial risk data of imported and exported frozen beverages from major global trading countries between 2015 and 2024, the distribution of non-compliance incidents across various categories of frozen beverages is as follows: ice cream 104 cases (67.10%), ice lolly 15 cases (9.68%), sweet ice 16 cases (10.32%), ice milk 16 cases (10.32%), ice frost 1 case (0.65%), and others 3 cases (1.94%). The comparison of microbial non-compliance in different categories of frozen beverages is shown in Table 4.

Further statistical analysis of the reasons for non-compliance in microbial indicators across different product categories is illustrated in Figure 4. Globally, the microbial contamination of ice cream products is particularly prominent. Among the 104 non-compliance incidents, the proportion of events exceeding the limit for coliforms reached 69.23%, while total plate count exceeded the limit in 12.50% of cases, and composite exceedance events accounted for 9.62%. A horizontal comparison with other categories of frozen beverages revealed that the primary reason for non-compliance in ice lolly, sweet ice, ice milk, and ice frost was also the exceedance of coliforms, with rates of 80.00%, 75.00%, 50.00%, and 100.00%, respectively. This underscores the significant role of coliforms non-compliance as a critical microbial risk across all types of frozen beverages. Notably, various foodborne pathogens were detected in ice cream products, including *L. monocytogenes* in 3 cases, *S. aureus* in 3 cases, *Salmonella* in 2 cases, and Norovirus in 1 case. The high risk associated with ice cream products provides a clear focus for inspection in global trade import and export checks.

### 3.5. Spatiotemporal Distribution of Microbial Safety Incidents in Frozen Beverages in the Chinese Market

A total of 903 incidents of non-compliance with microbial indicators in the safety supervision and sampling inspections of frozen beverages in the Chinese market from 2015 to 2024 were collected for this study. The distribution of these incidents by year is illustrated in Figure 5a. Notably, the year 2021 recorded the highest number of non-compliance incidents, reaching 143 cases, with the fourth quarter of 2021 alone accounting for 83 cases. In contrast, the year 2024 saw the fewest incidents, with only 38 cases reported throughout the year. The number of incidents for the remaining years is as follows: 124 cases in 2015, 59 cases in 2016, 114 cases in 2017, 101 cases in 2018, 105 cases in 2019, 87 cases in 2020, 59 cases in 2022, and 72 cases in 2023. Over the past decade, the annual distribution of non-compliance incidents in the safety supervision and sampling inspections of frozen beverages in the Chinese market has exhibited a fluctuating downward trend.

The comparison of microbial safety incidents in different regions of China is presented in Table 5. The geographical distribution of the origin of products involved in non-compliance incidents is shown in Figure 5b. Overall, the top five regions of origin are: Guangdong Province (131 cases, 14.52%), Liaoning Province (92 cases, 10.2%), Heilongjiang Province (89 cases, 9.87%), Henan Province (62 cases, 6.87%), and Shandong Province (54 cases, 5.99%). The regions with the lowest incidence were the Ningxia Hui Autonomous Region (1 case, 0.11%) and Qinghai Province (1 case, 0.11%).

Further analysis of the risk characteristics of frozen beverages in different regions of China reveals a significant geographical clustering of their origins, primarily concentrated in South China, Northeast China, and the Huang-Huai-Hai region. Specifically, the South China region is centered around Guangdong Province, while the Northeast region is formed by Liaoning, Heilongjiang, and Jilin Provinces. The Huang-Huai-Hai region is primarily represented by Henan and Shandong Provinces. It is noteworthy that this geographical clustering of non-compliance incidents aligns closely with the regional distribution of major manufacturers in the Chinese frozen beverage industry.

### 3.6. Microbial Safety Incident Contamination Spectrum of Frozen Beverages in the Chinese Market

The distribution of non-compliant microbial indicators in the safety supervision and sampling inspections of frozen beverages in the Chinese market from 2015 to 2024 is illustrated in Figure 6. The non-compliance rates of microbial indicators show a high degree of similarity with the testing results in global import and export trade. Within the Chinese market, incidents of excessive coliforms account for 41% of the non-compliance cases in frozen beverages, while incidents involving both coliforms and total plate count exceedance comprise 30%. Furthermore, incidents of total plate count exceedance represent 28.4% of the cases.

It is noteworthy that the proportion of non-compliance incidents involving various pathogenic microorganisms, including *L. monocytogenes*, is 0.6%, which is lower than the 6.4% rate of pathogenic bacteria non-compliance found in frozen beverages in global trade. Specifically, the rates for *L. monocytogenes* (0.4%), *S. aureus* (0.1%), and *Salmonella* (0.1%) are relatively low; however, the potential threat these pathogens pose to consumer health should not be overlooked.

### 3.7. Cluster Analysis of Microbial Safety Incidents in Frozen Beverage Market in China

Based on the safety supervision and sampling inspection data of frozen beverages in the Chinese market from 2015 to 2024, the comparison of microbial non-compliance in different categories of frozen beverages is shown in Table 6. The specific distribution of non-compliance incidents across various product categories is as follows: ice milk 401 cases (44.41%), ice lolly 310 cases (34.33%), sweet ice 71 cases (7.86%), ice cream 48 cases (5.32%), ice frost 46 cases (5.09%), edible ice 15 cases (1.66%), and others 12 cases (1.33%).

The statistical results are illustrated in Figure 7. Further analysis reveals that microbial contamination is particularly prominent in ice milk and ice lolly products. This contrasts sharply with the high-risk situation of ice cream products in global import and export trade, as the categories of non-compliant frozen beverages in the Chinese market are concentrated in ice milk and ice lolly products. Among the 401 non-compliant ice milk incidents, the proportion of events exceeding the standard for coliforms reached 44.14%, while the total plate count exceeded the standard in 24.19% of cases, and composite exceedance incidents accounted for 30.67%. In the 310 non-compliant ice lolly incidents, the proportion of events exceeding the standard for coliforms was as high as 39.35%, with total plate count exceedance in 28.06% of cases, and composite exceedance incidents comprising 32.58%. The distribution of pathogenic microorganisms exceeding standards exhibited heterogeneity across product categories: *L. monocytogenes* was detected in 4 cases of ice milk, *S. aureus* in 1 case of ice cream, and *Salmonella* in 1 case of sweet ice.

## 4. Discussion

This study systematically analyzes the spatiotemporal distribution patterns, contamination characteristics, and category-specific risk differences in microbial safety incidents related to frozen beverages globally and in China from 2015 to 2024. By revealing key risk factors from a three-dimensional perspective encompassing geographic spatiotemporal factors, microbial indicators, and product categories, this research provides a scientific basis for the formulation of targeted prevention and control strategies.

Microbial contamination of frozen beverages exhibits significant global regional clustering. The Southeast Asian region, particularly Malaysia (16.13%) and Thailand (11.61%), presents notable risks, while Western Europe shows lower risks. This may be attributed to the tropical environment of Southeast Asia, where high temperatures and humidity provide suitable conditions for microbial proliferation, with Malaysia’s average daily temperature ranging from 21.5 to 30.5 °C [18]. Additionally, differences in the adequacy of cold chain infrastructure lead to varying degrees of temperature fluctuation risks during storage and transportation, thereby affecting the succession patterns of microbial communities [19]. This differentiation is closely related to regional sanitary conditions: developing countries generally face weak hygiene management in production, as evidenced by a *Salmonella* prevalence of up to 23.81% in ice cream from Nigeria [20], and the detection of *S. aureus* (5.5%) and *Salmonella* (3.6%) in commercially available ice cream in Turkey [21]. Notably, risk heterogeneity also exists within countries—high-risk areas in China are concentrated in South China (Guangdong 14.52%), Northeast China (Liaoning 10.20%, Heilongjiang 9.87%), and the Huang-Huai-Hai region (Henan 6.87%, Shandong 5.99%). This “globally widespread—locally clustered” characteristic indicates that high-temperature and high-humidity climates, levels of sanitary infrastructure, and the enforcement of production process standards are core driving factors affecting the microbial safety of frozen beverages.

Analysis of microbial indicators further reveals common risks: the presence of coliforms above acceptable levels constitutes a systemic threat to the safety of frozen beverages. Coliforms are indicators, not direct pathogens, but exceeding limits still signals poor hygiene and potential risk. This phenomenon accounts for as much as 67.7% of non-compliance incidents in global import and export trade, with the Chinese market reaching 41%, far exceeding other indicators. It is noteworthy that the current industrial production system has established mature pasteurization processes, yet instances of contamination persist. Evidence indicates that psychrophiles bacteria are one of the key factors contributing to this contradiction: the heat-resistant proteases and lipases they produce can withstand pasteurization and even ultra-high temperature treatments, retaining 60–70% and 30–40% of their enzymatic activity, respectively. This leads to continuous spoilage of products during storage and transportation. Additionally, these microbial populations can rapidly proliferate under refrigeration conditions, becoming dominant microorganisms (accounting for up to 90%), resulting in significant economic losses [22]. Further research shows that psychrophiles Bacillus cereus sensu stricto has a high detection rate in raw milk and processing environments (with 13% positive samples), and 91.8% of the strains exhibit proteolytic activity. The heat-resistant protease genes they carried can remain active even after pasteurization, severely impacting product quality and safety [23]. The secondary growth of psychrophiles bacteria during low-temperature production may be a key contributing factor; these microorganisms can initiate metabolic compensation mechanisms in environments ranging from −1.5 °C to 4 °C, rapidly proliferating under low-temperature conditions by synthesizing cold shock proteins (CSPs) and accumulating compatible solutes [24]. Cases such as 37% of retail ice samples in the U.S. exceeding coliforms limits and detecting *Salmonella* [25], and 56% of Turkish ice cream being contaminated with *E. coli* [26], corroborate this issue. Given that exceeding these limits can lead to potential possibility of gastrointestinal diseases and even food poisoning, both China and the EU have strict standards (≤10^2^ CFU/g) [27,28]. This necessitates the development of a predictive microbiology model-based software system that covers the entire supply chain of raw materials, processing, and storage and transportation, enabling precise control of production processes through dynamic simulations of the growth boundary conditions of psychrophiles bacteria (temperature/pH/water activity).

Although pathogenic microorganisms account for only 6.4% of global non-compliance incidents (0.6% in China), their health risks should not be underestimated. *L. monocytogenes* (2.6% globally/0.4% in China), *S. aureus* (1.9%/0.1%), *Salmonella* (1.3%/0.1%), and Norovirus (0.6%) constitute the main pathogenic spectrum, with *L. monocytogenes* associated with a mortality rate of up to 30% from listeriosis [29]. The risk stems not only from high pathogenicity but also from the protective biofilm effect of pipeline systems in the production process of frozen beverages. Research indicates that even in a cold chain environment of 6 °C, *Pseudomonas* species can form biofilms on stainless steel and rubber surfaces. Within these biofilms, the bacterial cells exhibit significantly higher protease activity—averaging approximately 50 times that of planktonic cells, with certain strains reaching up to 190 times—thereby greatly enhancing their ability to cause spoilage in dairy products at low temperatures [30]. In a 2015 incident in the U.S. involving contaminated ice cream, the detection rate of *L. monocytogenes* in production lines reached 99% [31], highlighting the persistent colonization ability of these pathogens in processing environments. The research conducted by Kumari and Sarkar on dairy products in India, which reported that the contamination rate of *Bacillus cereus* in ice cream can reach up to 40%. The contamination levels ranged from 10^2^ to 10^8^ CFU/mL, and over 70% of the isolated strains retained their biofilm-forming ability at 4 °C. The authors indicated that the biofilms formed by these strains on processing equipment are the primary source of recurrent contamination in products, as they exhibit greater resistance to cleaning and disinfection measures, posing a persistent threat to quality and safety [32]. Biofilms in ice cream factories can harbor pathogens such as *Listeria* and *Shigella* [33]. Furthermore, the strong adhesion of *L. monocytogenes* to stainless steel exacerbates the risk of cross-contamination [34,35]. Therefore, it is crucial to focus on preventing biofilm formation and to develop a CIP (Clean-In-Place) system targeting pipeline systems, optimizing cleaning agent formulations and fluid dynamics design to achieve thorough removal of biofilms.

There is significant category-specific differentiation in the risks associated with frozen beverages. Ice cream issues are most prominent in global trade, accounting for 67.10% of non-compliance incidents, with long supply chains potentially amplifying contamination risks. However, the Chinese market exhibits different characteristics: ice milk (44.41%) and ice lolly (34.33%) emerge as the primary risk carriers, while emerging categories such as edible ice also warrant attention (1.66%). This may be due to the fact that products are typically produced by small factories with varying formulations. Mechanistic studies indicate that the coliforms exceedance rate in ice cream samples from Chinese dining establishments reaches 15.69%, with contamination levels significantly higher at the dining end compared to the retail end [36], confirming the key role of recontamination. *Salmonella* or *Listeria* contamination often arises from exposure to contaminated environments post-processing [37], with factors such as raw material defects, poor water quality, and temperature fluctuations during storage and transportation synergistically exacerbating risks [38,39]. This category differentiation necessitates the establishment of targeted prevention and control strategies: for high-risk categories (global ice cream, Chinese ice milk/ice lolly), enhanced monitoring of biofilms and management of recontamination points is essential.

Based on the aforementioned research, the following specific recommendations are proposed: First, the HACCP system can be further optimized by analyzing regional and category-specific risk characteristics. For products originating from Southeast Asia and high-incidence areas in China such as South China and Northeast China, it is advisable to strengthen the monitoring of raw material acceptance and the post-pasteurization process, with particular emphasis on the cleaning and sanitation of pipelines after heat treatment. This will effectively prevent secondary contamination by psychrophiles pathogens and the formation of biofilms. For high-risk categories (such as ice cream in global trade and ice milk/ice lolly in the Chinese market), stricter microbial limits should be established at critical control points, and sampling frequency should be increased to enhance daily monitoring of coliforms and specific pathogenic microorganisms.

Secondly, efforts should be made to promote international coordination and mutual recognition of microbial limit standards. This study indicates that exceeding coliforms limits is a prominent global issue (accounting for 67.7% of global incidents, with 41% in China). It is recommended to advocate for the harmonization of limit standards and testing methods for hygiene indicator bacteria (such as coliforms and total plate count) at the international level, in order to reduce trade barriers and regulatory loopholes caused by standard discrepancies. Additionally, there should be a focus on unifying limit standards for pathogens such as *L. monocytogenes* and *Salmonella* to enhance the safety of frozen beverages globally.

Furthermore, it is essential to strengthen temperature control throughout the entire supply chain and ensure compliance with cold chain management. The promotion of intelligent time-temperature monitoring technology will enable real-time monitoring and early warning of temperature fluctuations throughout the production to consumption process, thereby minimizing the proliferation of psychrophiles bacteria and secondary contamination.

Finally, it is recommended to establish a multinational data-sharing and risk warning collaborative platform to facilitate efficient communication of risk information between regulatory agencies and the food industry, supporting evidence-based precise regulation and supply chain risk intervention.

This study has several limitations. First, the analysis relies entirely on official reports and data published by various countries, which may be influenced by differences in regulatory intensity, inspection frequency, and reporting transparency, leading to potential underreporting or reporting bias. Second, inconsistencies in microbial testing items, limit standards, and sampling methods across different countries or regions may affect the direct comparability of the data. Furthermore, due to the inability to access internal monitoring records and non-public voluntary recall information from companies, the actual number of contamination incidents may be higher than the statistics reported in this study. Future research could further reduce uncertainty and enhance the reliability of cross-regional risk comparisons by integrating multi-source data (such as industry self-inspection reports and laboratory monitoring network data) and employing standardized risk adjustment methods.

## 5. Conclusions

This study fills a gap in the multidimensional analysis of microbial risks in frozen beverages and points to future research directions: firstly, it is necessary to construct differentiated risk assessment models based on product categories, integrating data on raw material characteristics, process parameters, and supply chain logistics; secondly, it is crucial to promote international mutual recognition of hygiene indicators and pathogenic bacteria limits to reduce cross-border trade risks. In the current context of global circulation, the complexity of microbial contamination in frozen beverages urgently requires interdisciplinary solutions. Future efforts should integrate microbiomics, materials surface engineering, and intelligent monitoring technologies to block contamination pathways at the source and ensure food safety.

## Figures and Tables

**Figure 1 foods-14-03238-f001:**
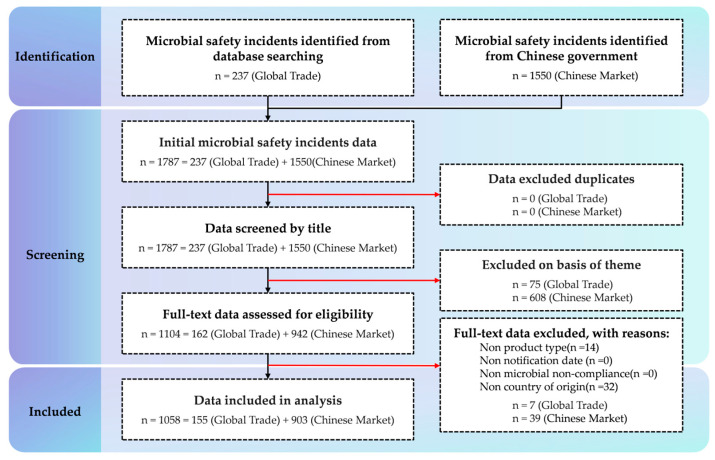
Flow diagram according to the PRISMA method.

**Figure 2 foods-14-03238-f002:**
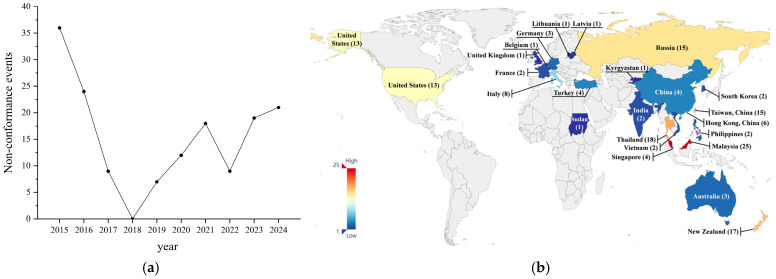
(**a**) Time distribution of microbial safety incidents in frozen beverage imports and exports among major global trading countries (2015–2024). (**b**) Heat map of microbial safety incidents in frozen beverages corresponding to the countries of origin of major global trading nations (2015–2024).

**Figure 3 foods-14-03238-f003:**
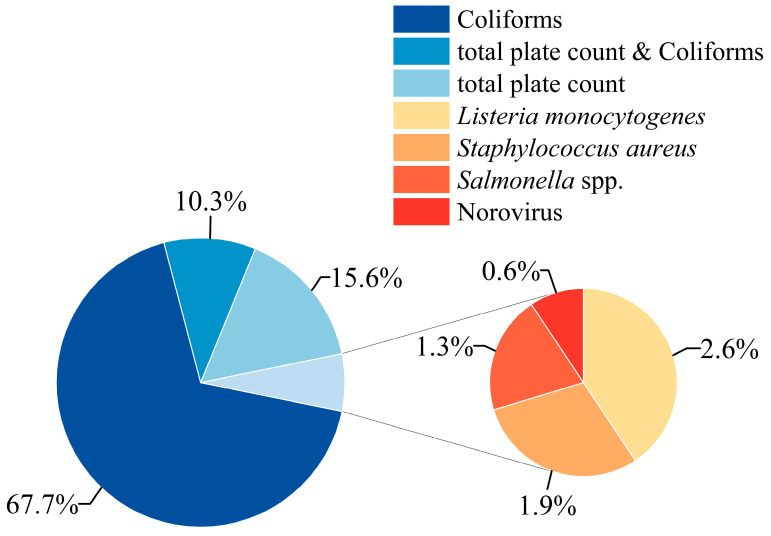
Microbial safety incidents and non-compliance rates of microbial indicators in frozen beverage imports and exports of major global trading countries (2015–2024).

**Figure 4 foods-14-03238-f004:**
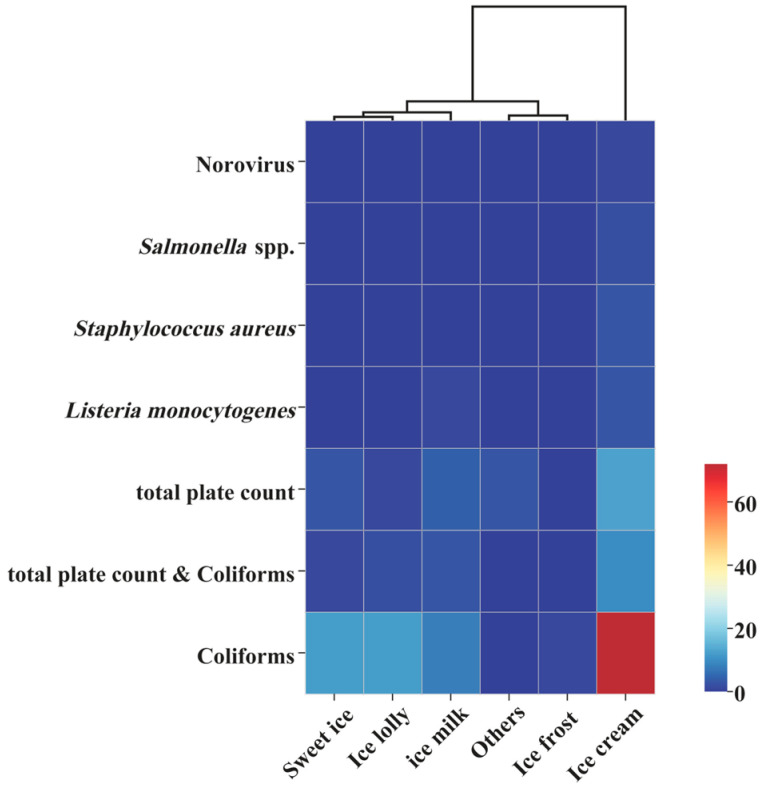
Cluster analysis of microbial safety incidents in frozen beverage imports and exports among major global trading countries (2015–2024).

**Figure 5 foods-14-03238-f005:**
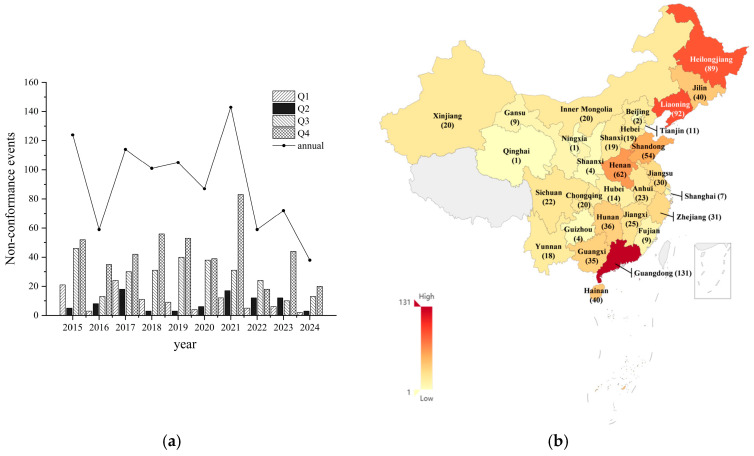
(**a**) Time Distribution of Microbial Safety Incidents in Frozen Beverage Market in China (2015–2024). (**b**) Heat Map of Microbial Safety Incidents in Frozen Beverages Market in China Corresponding to Place of Origin (2015–2024).

**Figure 6 foods-14-03238-f006:**
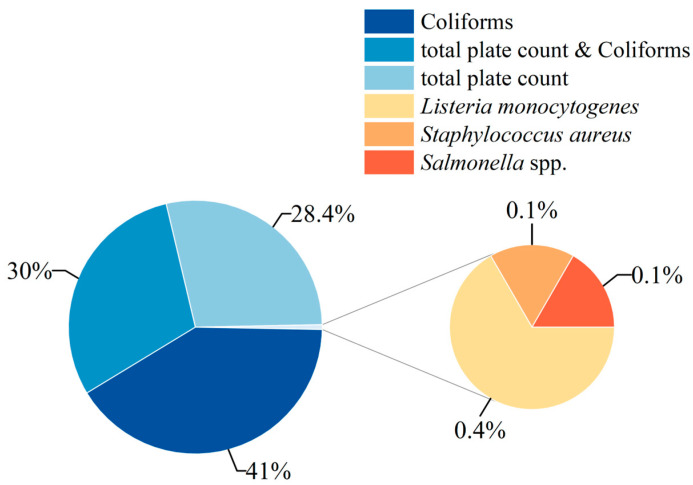
Microbial Safety Incidents and Non-compliance Rates of Microbial Indicators in Frozen Beverage Market in China (2015–2024).

**Figure 7 foods-14-03238-f007:**
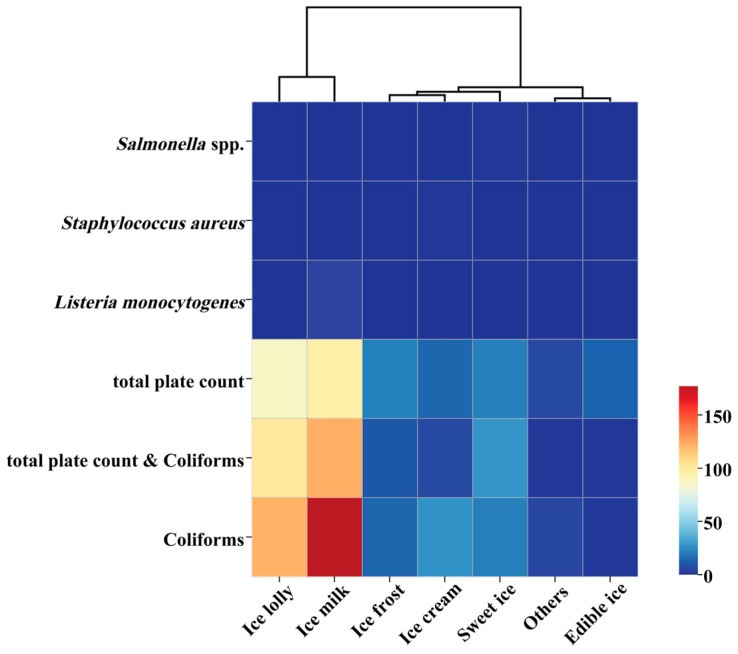
Cluster Analysis of Microbial Safety Incidents in Frozen Beverage Market in China (2015–2024).

**Table 1 foods-14-03238-t001:** Data Sources and Website Link.

Data Sources	Web Section	Website Link
Rapid Alert System for Food and Feed (RASFF)	/	https://webgate.ec.europa.eu/rasff-window/screen/search (accessed on 10 September 2025)
U.S. Food and Drug Administration	Import Refusal Section, Import Alert Section	https://www.fda.gov/food/guidance-regulation-food-and-dietary-supplements/food-imports-exports (accessed on 10 September 2025)
Ministry of Health, Labour and Welfare of Japan	Section on Violations in Food Monitoring Operations	https://www.mhlw.go.jp/stf/seisakunitsuite/bunya/kenkou_iryou/shokuhin/yunyu_kanshi/index.html (accessed on 10 September 2025)
Korea Food and Drug Administration	Imported Food Non-compliance Sector	https://www.mfds.go.kr/eng/wpge/m_11/de011002l001.do
Department of Agriculture, Fisheries and Forestry of Australia	Inspection and Testing of Imported Food Sector	https://bicon.agriculture.gov.au/BiconWeb4.0/Home/Notice (accessed on 10 September 2025)
New Zealand Food Safety Authority	Food Recall and Complaints Division	https://www.mpi.govt.nz/food-safety-home/food-recalls-and-complaints/ (accessed on 10 September 2025)
General Administration of Customs of China	Information on Prohibited Food and Cosmetics	http://jckspj.customs.gov.cn/spj/xxfw39/fxyj47/4677516/index.html (accessed on 10 September 2025)
State Administration for Market Regulation and Provincial Bureaus	/	https://www.samr.gov.cn/ (accessed on 10 September 2025)
Food Safety Center of Hong Kong	Food Alert	https://www.cfs.gov.hk/tc_chi/whatsnew/whatsnew_fa/whatsnew_fa.html (accessed on 10 September 2025)
Taiwan Food and Drug Administration	Non-compliance of Food Information in Import Inspection	https://www.fda.gov.tw/TC/index.aspx (accessed on 10 September 2025)

**Table 2 foods-14-03238-t002:** Included data characteristics.

Data Characteristics		Number of Data
Incidents country & region	Southeast Asia	51
	Other Asian region	34
	Europe	32
	Africa	1
	America	13
	Oceania	20
	South China	215
	North China	71
	East China	63
	Northeast China	221
	Southwest China	64
	Northwest China	35
	Central China	73
	Huang-Huai-Hai region	146
Beverage category	ice cream	152
	ice milk	417
	ice frost	47
	ice lolly	325
	sweet ice	87
	edible ice	15
Non-compliance items	Coliforms	472
	Total plate count	284
	Coliforms & total plate count	286
	*L. monocytogenes*	8
	*S. aureus*	4
	*Salmonella*	3
	Norovirus	1

**Table 3 foods-14-03238-t003:** Comparison of microbial safety incidents across different regions worldwide. The bold statistical values refer to the overall significance for incidents data across different regions.

Region	Countries/Regions	Number of Data
Southeast Asia	Malaysia, Thailand, Singapore, Vietnam, Philippines	51
Other Asia region	China Taiwan, China Hong Kong, China, India, South Korea, Turkey, Kyrgyzstan	34
Europe	Russia, Italy, Germany, France, Latvia, Lithuania, Belgium	32
Africa	Sudan	1
America	United States	13
Oceania	New Zealand, Australia	20
	**X^2^**	**61.623**
	** *p* **	**<0.001**

**Table 4 foods-14-03238-t004:** Comparison of microbial non-compliance in different categories of frozen beverages in global import and export trade. The bold statistical values refer to the overall significance for incidents data. N.s. indicates non significant differences.

Non-Compliant Items	Categories of Frozen Beverages						X^2^	*p*
	Ice Cream	Ice Lolly	Sweet Ice	Ice Milk	Ice Frost	Others		
Norovirus	1	0	0	0	0	0	**n.s**	
*Salmonella* spp.	2	0	0	0	0	0	**n.s**	
*S. aureus*	3	0	0	0	0	0	**n.s**	
*L. monocytogenes*	3	0	0	1	0	0	**1.000**	**0.317**
total plate count	13	1	3	4	0	3	**18.500**	**<0.001**
total plate count & Coliforms	10	2	1	3	0	0	**12.500**	**0.006**
Coliforms	72	12	12	8	1	0	**158.667**	**<0.001**
**X^2^**	**264.577**	**14.800**	**12.875**	**6.500**	**n.s**	**n.s**		
** *p* **	**<0.001**	**<0.001**	**0.002**	**0.090**				

**Table 5 foods-14-03238-t005:** Comparison of microbial safety incidents across different regions in china. The bold statistical values refer to the overall significance for incidents data across different regions.

Region	Provinces	Number of Data
Northeast China	Liaoning, Heilongjiang, Jilin	221
South China	Guangdong, Hainan, Guangxi, Fujian	215
Huang-Huai-Hai region	Henan, Shandong, Jiangsu	146
Central China	Hunan, Anhui, Hubei	73
North China	Inner Mongolia, Shanxi, Hebei, Tianjin, Beijing	71
Southwest China	Sichuan, Chongqing, Yunnan, Guizhou	64
East China	Zhejiang, Jiangxi, Shanghai	63
Northwest China	Xinjiang, Gansu, Shaanxi, Qinghai, Ningxia	35
	**X^2^**	**337.60**
	** *p* **	**<0.001**

**Table 6 foods-14-03238-t006:** Comparison of microbial non-compliance in different categories of frozen beverages in Chinese market. The bold statistical values refer to the overall significance for incidents data. n.s. indicates non significant differences.

Non-Compliant Items	Categories of Frozen Beverages							X^2^	*p*
	Ice Milk	Ice Lolly	Sweet Ice	Ice Cream	Ice Frost	Edible Ice	Others		
*Salmonella* spp.	0	0	1	0	0	0	0	**n.s**	
*S. aureus*	0	0	0	1	0	0	0	**n.s**	
*L. monocytogenes*	4	0	0	0	0	0	0	**n.s**	
total plate count	97	87	21	14	22	13	6	**232.800**	**<0.001**
total plate count & Coliforms	123	101	28	6	10	1	1	**410.607**	**<0.001**
Coliforms	177	122	21	27	14	1	5	**540.997**	**<0.001**
**X^2^**	**156.436**	**6.006**	**22.915**	**32.167**	**4.870**	**19.200**	**3.500**		
** *p* **	**<0.001**	**0.050**	**<0.001**	**<0.001**	**0.088**	**<0.001**	**0.174**		

## Data Availability

The original contributions presented in the study are included in the article, further inquiries can be directed to the corresponding author.

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
