# Peer review of "Analysis of Global Microbial Safety Incidents in Frozen Beverages from 2015 to 2024"

_foods, 2025, doi:10.3390/foods14183238_

Round 1

Reviewer 1 Report

Comments and Suggestions for Authors

The manuscript addresses an important and timely topic in food safety; however, the novelty of the work compared to existing literature should be highlighted more explicitly in the introduction.

  1. The abstract provides a comprehensive overview, but it should end with a stronger concluding statement on the study’s practical implications 
  2. The introduction provides a good background but could benefit from a more direct statement of research gaps, particularly why frozen beverages have been overlooked in systematic microbial risk assessments.
  3. Provide more justification for choosing the 2015–2024 period, and clarify if data from earlier years were unavailable or excluded.
  4. The methodology is described clearly; however, more detail is needed on the search strategy and the criteria for including and excluding incidents.
  5. Explain how potential duplicate reports across regulatory agencies were handled to avoid double-counting.
  6. As the authors mentioned that “this study systematically collected and analyzed 155 records of microbial safety,” the methodology should include a PRISMA flow diagram and a table detailing study characteristics.
  7. The results are comprehensive but somewhat descriptive; adding statistical comparisons (e.g., chi-square tests for differences between regions or categories) would strengthen the findings.
  8. Some claims (e.g., psychrophilic bacterial growth and biofilm persistence) need stronger citation support or experimental evidence in the discussion section.
  9. Provide a clearer comparison with previous global studies on ice cream and frozen desserts to highlight the novel aspects of this research.
  10. The conclusions should emphasize how this analysis can guide specific interventions (e.g., HACCP improvements, international harmonization of microbial limits).
  11. Consider adding a short section on limitations (e.g., reliance on official reports, underreporting bias, differences in regulatory standards across countries)
Comments on the Quality of English Language

It can be improved further

Reviewer 2 Report

Comments and Suggestions for Authors

The microbiological safety of ice cream and related products has long been a subject of research; however, any study addressing global trends provides an important contribution. This manuscript is of interest both nationally and internationally, as it highlights common challenges that represent key microbiological issues in the production of ice cream and similar products. The study emphasizes the spatiotemporal distribution of microbial safety incidents at both global and national levels. The research aim is well defined and has been largely achieved; nevertheless, certain improvements are required. The overall study design is appropriate and, following revisions, will be sufficient to adequately address the stated aim

Reviewer 3 Report

Comments and Suggestions for Authors

This manuscript presents a systematic analysis of microbial safety incidents in frozen beverages between 2015 and 2024, drawing on global regulatory databases (155 incidents) and Chinese domestic surveillance data (903 incidents). The authors identify the presence of coliforms as the predominant issue both internationally and in China, while pathogenic microorganisms (Listeria monocytogenes, Salmonella spp., Staphylococcus aureus, and Norovirus) are much less frequent but still of concern. The study also highlights product-specific differences: ice cream is most problematic globally, whereas ice milk and ice lollies dominate the Chinese non-compliance data. The strength of this work lies in the volume of the dataset, the integration of global and national perspectives, and the systematic statistical breakdown by region and product type. These findings provide useful baseline information for risk assessment and regulatory strategies.

Round 2

Reviewer 1 Report

Comments and Suggestions for Authors

The length of abstract is high, it must be put in accordance with journal guidelines

Overall correction is good with few reading mistakes like diagrams were performed by origin 2021. It should be rather prepared as diagrams are not something that is performed

The inclusion and exclusion criteria heading could be written separately rather than combined

Reviewer 2 Report

Comments and Suggestions for Authors

Revisions have been made to the manuscript according to the reviewer's requests, however a few minor changes still need to be made

LInes 125-126. Delete sentence Frozen beverages are sometimes referred to as frozen
drinks or frozen desserts.

Conclusion section provides valuable recommendations, however they are to detailed which makes Conclusion section unnecessarily long. These recommendations should be moved to the end of Discussion  section (before study limitations), while only the paragraph from line 579 to 588 should remain in Conclusion section.
